# Vegetables and fruits retailers in two urban areas of Bangladesh: Disruption due to COVID– 19 and implications for NCDs

Md. Nazmul Hossain[1,2], Md. Saiful Islam [1,2], S. M. Abdullah [1,2,3]*, Syed Mahbubul Alam [4], Rumana Huque[1,2]

**1** Department of Economics, University of Dhaka, Dhaka, Bangladesh, **2** Research and Development, ARK Foundation, Dhaka, Bangladesh, **3** Department of Health Sciences, University of York, York, United Kingdom, **4** Center for Law and Policy Affairs (CLPA), Dhaka, Bangladesh

* smabdullah@du.ac.bd

**Data Availability Statement:** All relevant data are within the paper and its Supporting Information files.

## Abstract

Bangladesh is experiencing an increasing prevalence of diet-related non-communicable diseases (NCDs). Considering daily total requirement of 5 servings as minimum recommended amount, 95.7% of people do not consume adequate fruit or vegetables on an average day in the country. Imposition of lockdown during COVID-19 created disturbance in fresh fruits and vegetable production and their retailing. This incident can make these dietary products less affordable by stimulating price and trigger NCDs. However, little is known about the supply chain actors of healthy foods such as vegetables and fruits in urban areas, and how they were affected due to pandemic. Aiming toward the impact of COVID–19 on the business practices and outcomes for the vegetables and fruits retailers in Bangladesh, a survey of 1,319 retailers was conducted in two urban areas, namely Dhaka and Manikganj from September 2021 to October 2021. To comprehend the impact of COVID-19 on the profit margin of the retailers and on the percentage change in sales, a logistic and an Ordinary Least Squares (OLS) regression were estimated. Significant difference in the weekly business days and daily business operations was observed. The average daily sales were estimated to have a 42% reduction in comparison to pre-COVID level. The daily average profit margin on sales was reportedly reduced to 17% from an average level of 21% in the normal period. Nevertheless, this impact is estimated to be disproportionate to the product type and subject to business location. The probability of facing a reduction in profit margin is higher for the fruit sellers than the vegetable sellers. Contemplating the business location, the retailers in Manikganj (a small city) faced an average of 19 percentage points less reduction in their sales than those in Dhaka (a large city). Area-specific and product-specific intervention are required for minimizing the vulnerability of retailers of vegetables and fruits and ensuring smooth supply of fruits and vegetables and increasing their uptake to combat diet related NCD.

**Funding:** This study is part of the research project titled "Fiscal and regulatory mechanisms for promoting healthy diet in urban Bangladesh: A Mixed Method Supply Chain Study" implemented by ARK Foundation, Dhaka, Bangladesh, with financial support from International Development Research Centre (IDRC), Canada. The project bears the grant number as 109264-001 and has implementation period from 1st January 2020 to 31st December 2022. The funders had no role in study design, data collection and analysis, decision to publish, or preparation of the manuscript.

**Competing interests:** The authors have declared that no competing interests exist.

## 1. Introduction

Bangladesh, a lower middle-income country in Southeast Asia, is experiencing an increasing prevalence of diet-related non-communicable diseases (NCDs). It is estimated that about 580,000 deaths are caused by NCD annually, representing more than 67% of all deaths in Bangladesh [1], and unhealthy lifestyles and diets play a leading role in this epidemic. Bangladesh is facing the challenges of huge childhood stunting, micronutrient deficiencies including anemia among women and children, and emerging over-nutrition (overweight and obesity) [2]. The prevalence of anemia is more than 50% among pregnant women [3]. In Bangladesh, overall daily per capita consumption of fruits was 1.7 servings and of vegetables was 2.3 servings [4]. Considering the daily total requirement of 5 servings as the minimum recommended amount [5], 95.7% of people do not consume an adequate amount of fruits or vegetables on an average day [4]. NCDs cause a huge direct impact on financial vulnerability, particularly for the poor, and an indirect impact on the economy [4]. The ongoing shift in disease burden and risk factors, therefore, warrants re-prioritization of public health interventions in Bangladesh. Food stamp program or supplementary nutrition program is widely applied in developed countries to make fruits and vegetables affordable for low-income households. However, a healthy diet incentive program is not much prominent in underdeveloped and developing countries and Bangladesh is not an exception. Additionally, the supply-side intervention to enhance the uptake of such diet is even scarce.

Households reduced their consumption of necessary commodities due to the adverse impact of COVID-19 and the lower income households were affected mostly compared to those from the richest quintile [6]. The imposition of lockdown and maintenance of social distancing created a disturbance in fresh fruits and vegetable production and their retailing [7]. Both the demand and supply side of the fruits and vegetable industry are disrupted due to the COVID–19 pandemic, but the direction and magnitude of the impact are ambiguous. COVID– 19, which has left none of the sectors untouched around the world, affected the dietary supply chain and hence has implications for NCD prevalence. Besides the Wet Market (WM) and Super Shops (SUP), vegetables and fruits in the cities of lower and lower-middle-income countries are mostly sold by Street Shops (SS) and Mobile Vendors (MV). Fresh fruits and vegetable vendors play an important role in providing these dietary foods at a lower price, especially to the low income and vulnerable populations [8]. A comparison between the pre and post pandemic time reveals that, of the identified vendors before COVID–19, 35% of the vendors were absent or closed business during the pandemic [7]. Such disruption caused by unprecedented COVID–19 on the important actor of the supply chain i.e. retailers, contains severe implications for healthy diet intake for low-income people in Bangladesh. This incident has the potential to stimulate prices and make important dietary products less affordable and put them further below the recommended daily intake and consequently, will trigger NCDs.

Considering the increasing prevalence of NCDs, little is known about the supply chain actors of healthy foods such as vegetables and fruits in urban areas and the impact of fiscal policies and regulatory measures in Bangladesh. This research conducted a survey among the vegetables and fruits retailers in Bangladesh with a general objective to explore the barriers and facilitators for promoting vegetables and fruit intake and to identify the gap in required fiscal and regulatory policy support. There is a lack of evidence in Bangladesh on the impact of COVID-19 on the fruits and vegetable retailers in terms of the magnitude of profit margin, daily sales, and business operation. Specifically, the current analysis aimed towards scrutinizing the impact of COVID–19 on the business practices and outcomes of vegetables and fruits retailers in Bangladesh. Functioning of the value chain of perishable goods were highly affected due to the imposition of countrywide lockdown [9, 10].Because of this downturn many people

involved in this sector lost their job not only in Bangladesh but also in other countries [11]. The research evidence revealed that there was a significant and negative impact of COVID-19 on the prices of fruits and vegetables [12]. The existing pieces of literature mainly focused on the COVID-19 disruption and the corresponding implication on food security. This paper contributes to the current literature in terms of examining the direct impact of COVID-19 disruption on the fresh fruits and vegetables retailers in the urban areas of Bangladesh.

The research objective was met answering the questions as follows, were the fruits and vegetables retailers being affected by COVID-19?, did the impact vary with retailers characteristics? and what was the magnitude of impact compared to the pre-COVID level in terms of daily sales, profit margin, business operation, etc? The structure of this paper contains introduction in section 1, methodology of the paper in section 2, descriptive and econometric results are discussed in section 3 and finally the discussion based on the results and conclusions are presented in section 4.

## 2. Materials and methods

### 2.1 Study design

This research is part of a sequential mixed-methods study in Bangladesh for promoting healthy diet. The study had three specific interconnected components: i) Point of Sale (POS) vendors survey; ii) In-depth interviews of vegetables and fruits supply chain actors; iii) Key Informant Interviews (KII) with policymakers and other relevant stakeholders. The quantitative component (POS survey) was used to design and developed the study tools and analytical aspects of qualitative components (in-depth interviews and KIIs). The study has an implementation span from January 2020 to December 2022. Exploiting the quantitative component i.e., POS survey data the current research output is developed.

### 2.2 Study sites and sampling frame

A questionnaire-based face-to-face survey of the vegetables and fruit retailers was conducted in two urban areas–Dhaka, a mega city, and Manikganj, a medium-sized city in Bangladesh. The survey areas were purposively selected based on the size of the city corporation. Inside Dhaka city corporations, 6 thanas (administrative sub-districts)–Badda, Lalbagh, Pallabi, Tejgaon, Mohammadpur, and Uttara–were randomly selected and inside Manikganj district, 2 thanas (administrative sub-districts)–Manikganj Sadar and Singair were selected to conduct the survey. Since an official list of fruits and vegetable retailers was unavailable, to find their number and location, a geo-mapping of markets and sellers was conducted before going to the actual field survey.

Fig 1 contains the map of the survey areas and S1 Table contains the sampling frame along with the distribution of vegetables and fruit retailers in two study areas against the type of market. As S1 Table shows, in Dhaka city, a total of 111 vegetables and fruit markets were geo mapped in the 6 study sub-districts. The distribution of WM and SUP was 81 against 30. In total 3,466 vegetables and fruit retailers were identified in these markets. Among them 2,166 were selling only vegetables, 1,242 were selling only fruits and 44 were selling both. Conversely in Manikganj entirely 16 WM were mapped which consisted of 633 retailers in the 2 study sub-districts. Against 373 only vegetable retailers, there were 245 exclusive fruit retailers and the rest 15 were selling both. In Manikganj no SUP was found during mapping. Besides WM retailers of fruits and vegetables, in both areas, street retailers were prominent. Nevertheless, MV for these foods are noticeably prominent in Dhaka city and near insignificant in Manikganj.

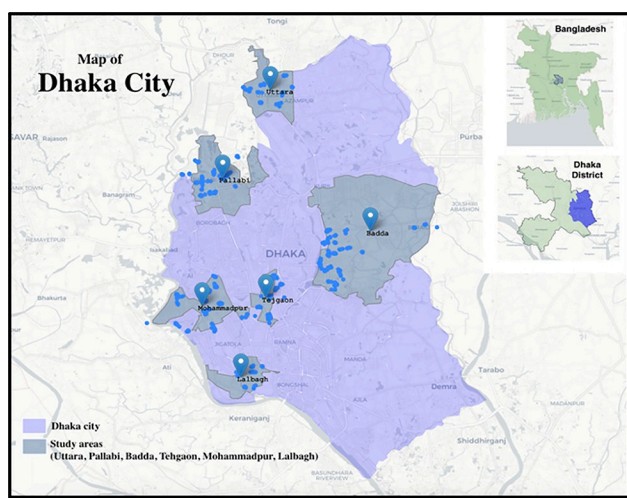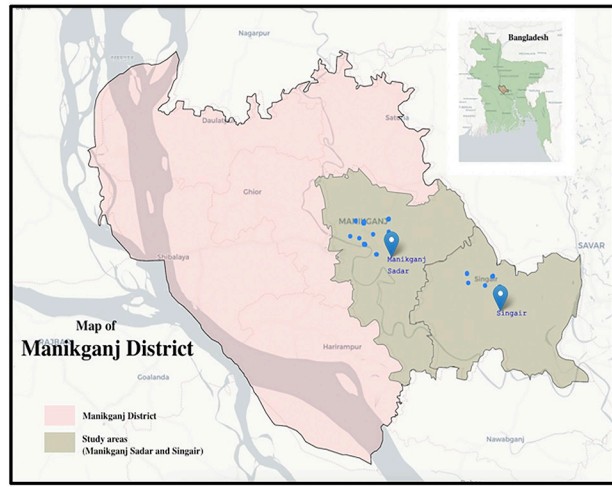

**Fig 1. The survey areas in Dhaka city and Manikganj district of Bangladesh.**

## 2.3 Sampling strategy, survey participants, and sample size

The survey used a mixed strategy with a composite sampling approach. Followed by a purposive selection of two study sites, random selection was made for selecting the subdistricts. These subdistricts were the Primary Sampling Unit (PSU). The smaller administrative areas named "ward" under the subdistricts in each study site were considered for enumeration. All WM, SS, and SUP were mapped under each selected enumeration area from PSU. The survey participants were fruits and vegetables retailers doing their business as sellers at WM, SS and MV. Systematic simple random sampling was followed to select them from those markets. More specifically, one-third of the retailers in each study site was surveyed by approaching every $k$th retailer where $k = 3$. The non-response rate was around 11%. The sample size consisted of 1,319 vegetable and fruits POS retailers; 1,121 were from Dhaka and the rest 198 were from Manikganj. Fig 2 contains the distribution of samples against the type of market and food. The sampling design idea is adopted from Khan et. al (2020) [13]. Based on the mapping result, during the field survey, the trained field enumerators considered the prominent street for business activities in the enumeration block and walked around to select the SS. In the enumeration areas within Dhaka city, there were some streets locally known for the "morning/ evening market". These markets are temporary and the retailers perform business on the street with moveable vans and carts for a stipulated time every day. While preparing the sampling frame and surveying the SS, those were also considered. Regarding recruiting the MV the enumerators walked around the residential places of the enumeration areas and approached and surveyed the vendors at their first point of contact.

## 2.4 Field design and implementation, data collection tools, and data management

A consecutive 7 weeks field during September-October 2021 was performed by 6 enumeration teams consisting of 2 enumerators for each. At first, all 21 enumeration blocks from Dhaka were completed for the survey, followed by 18 enumeration blocks from Manikganj were approached. For gender parity, an equal number of male and female enumerators were appointed for constructing the team. Multiple rounds of training were completed to familiarize and make the enumerators compatible with the data collection tool. The questionnaire was

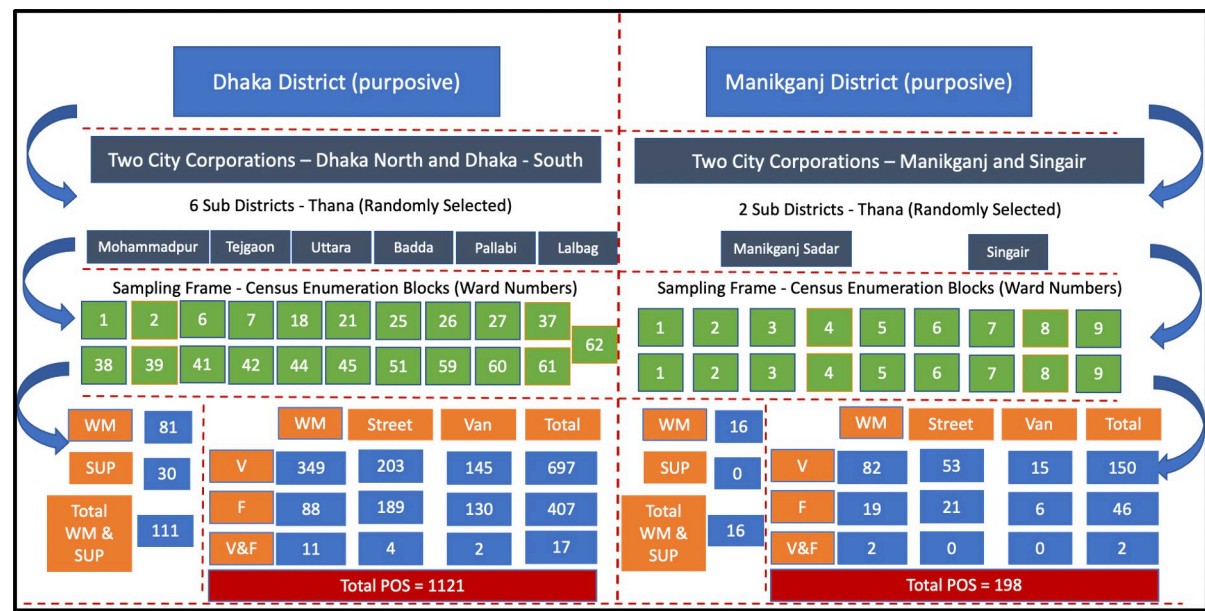

Note: The design idea is adopted from Khan et. al (2020)

**Fig 2. Sample of retailers in WM, SS, and MV of Dhaka and Manikganj.** Note: The design idea is adopted from Khan et. al (2020).

structured to obtain information about the business (revenue and profit) of the sellers, COVID-19 effect on business, costs of running a business and sellers' views about difficulties of doing business. Followed by paper-based development, the questionnaire was transferred to SurveyCTO [14], a mobile data collection platform, and the trained field enumerators used the SurveyCTO [14] application on their smartphones to conduct the survey. Two rounds of piloting of the data collection tool and hence validation was performed (one with the paper-based form and the other with the SurveyCTO application form). Before the final survey the essential refresher training also took place. There were 2 field supervisors and in the back office 2 exclusive data management personnel. The data management team checked for consistency and validation of the completed survey during each field implementation day using the SurveyCTO platform [14] and provided live feedback to the enumerators should there needed any clarification. The questionnaire and the survey were conducted in local language i.e., Bengali.

## 2.5 Ethical considerations

Ethical and community considerations are of prime importance for any study involving the collection of field information. There can arise insecurity issues from several aspects for the study participants. For instance, sellers in WM or in SUP might feel insecure as they need to answer queries on pricing, profit, and tax. Also, surveying them during business time might hamper their regular activities. Moreover, they might feel threatened further because of the potential disclosure of their personal and business secret. Our research team was considerate of all these aspects and requested the participants written consent during recruitment. While all potential participants were conversant about the study beforehand by the enumerator, they had the freedom of choice regarding participation. The rights of interview participants were given the highest priority and enumerators were specially instructed not to create any difficulty against facing any unwillingness of answering any question. The National Research Ethics Committee of the Bangladesh Medical Research Council (BMRC) (Ref: BMRC/NREC/2019–2022/983; Registration Number: 326 12 08 2020) provided the ethics clearance for the study.

## 2.6 Data analysis

Considering the quantitative nature, the data analysis used both descriptive and inferential statistical operations. In summary statistics, different measures of descriptive statistics such as average, median, and proportions from central tendency were used. Standard statistical tests i.e. $t$–test, $\chi^2$–test, $F$-test etc. were performed in due requirement for testing the hypothesis. Using the information on ownership of some basic assets such as Television, refrigerator, main materials of roof and walls of house etc., a wealth index has been constructed applying Principal Component Analysis (PCA) and then the wealth index distribution is divided into quintiles [15]. To discuss the effect of COVID-19, the retailers were asked to answer the same question corresponding two time periods. Pre–COVID–19 is referred to as the time period before the first official report of COVID–19 patient and the government announcement of a countrywide strict lockdown in mid-March 2020. In later period, the government announced a countrywide lockdown in multiple phases. Post—COVID– 19 is referred to as the time period after the government withdrew countrywide strict lockdown and just a week before the survey was conducted during September—October 2021.

In addition to the descriptive analysis, to understand the impact of COVID– 19 on the profit margin of the retailers and on the percentage change in sales compared to the pre–COVID level, two regression specifications are estimated. In specification 1, the following logistic regression is estimated,

$$p(y_i = 1) = \Phi(X'\beta + \varepsilon_i)$$

where the value of the dependent variable, $y_i$, is zero if the profit margin of the retailer remains unchanged and 1 if it is reduced from the pre–COVID level. In the above expression, $\varepsilon_i$, is the disturbance term and $\Phi(\cdot)$ is the cumulative logistic distribution. Here, $X$ is the vector of independent variables that include the main variables of interest which are product type (only vegetables, only fruits, and vegetables, fruits and others), retailer type (street or mobile vendor and wet market vendor), and business location (Dhaka and Manikganj). Besides, $X$ includes other variables like business license availability, age and experience of retailers, number of workers, and educational attainment which are used as control variables in the regression estimation.

In Specification 2, an Ordinary Least Squares (OLS) regression is applied to find the impact on the percentage change in average sales between the pre-COVID and early COVID period. The following specification was estimated:

$$y_i = X'\beta + \varepsilon_i$$

Where, $y_i = \frac{average\ sales\ during\ COVID - average\ sales\ pre\ COVID}{average\ sales\ pre\_COVID}$ and $X$ is the vector of independent variables as in logistic regression. Under the OLS regression, two different models are estimated. In Model 1, all the control variables of logistic regression are present while Model 2 is augmented with the square of age and business experience of the retailers. The percentage change in sales during COVID and pre–COVID is the outcome variable. Since almost all the values of the variable are negative, the lower the value of the variable the higher the reduction in sales in monetary terms. For diagnosing the robustness of the results in regression models, Variance Inflating Factor (VIF) [16, 17], Breusch-Pagan (BP) test of heteroscedasticity [18] and Ramsey's Regression Specification Error Test (RESET) [19] were performed. All statistical analysis was conducted using a statistical software version of STATA 17 [20].

## 3. Results

### 3.1 Socio-demographic and business profile of the vegetables and fruit retailers

S2 Table shows the socio-demographic and business characteristics of the vegetables and fruit retailers. Out of 1,319 retailers, 41.77% were from the WM while 35.63% were from SS and the rest 22.59% were MV. 64.22% of the retailers were involved only in vegetable retailing while 34.34% were doing fruit retailing. Among all respondents, 97.73% of the vendors were male. Most of the respondents' educational qualification was primary and below, 41.24% have only a primary level of education and 33.21% have had no formal education. The wealth distribution showed that only 4.47% of the respondents belong to highest wealth quintile, conversely 41.85% of respondents were in bottom two quintiles. The average age of the retailers was 38 years. Most of them belong to the age group of 31–40 years (34.42%). Considering business experience, 36.09% of the respondents had less than 5 years of experience, while 31.39% of the respondents had experience of 6–10 years. The average level of experience in the whole sample was around 10 years.

Most of the retailers (50.95%) had a very small (less than Bangladeshi Taka (BDT) 10,000) initial investment for the business. Self-investment was the sources of funds for 92.42% of the retailers as their initial cost of business, 35.33% borrowed from friends and relatives, and only 8.29% managed funds from NGOs, banks, or other financial institutions.

### 3.2 COVID–19 disruption on the business of the retailers

In Table 1, the impact of COVID-19 on the sales, profit, business operations, and other factors of the retail business owners is presented. Including the owner, on average 1.4 people were engaged in the business before the COVID-19 pandemic (February 2020) and this was reduced to 1.33 in the week before the survey (August 2021). This falloff in the business was statistically significant (two sample mean test, $p<0.05$), indicating that during the COVID–19 lockdown of multiple phases in the year 2020 and 2021, employment opportunity reduced in vegetables and fruit retailing business. During the lockdown period of COVID–19 in 2020 and 2021, consumers started to buy fruits and vegetables from via phone call where customers would call the retailers to place the order, and they would deliver the order. The proportion of retailers selling fruits and vegetables online (via phone call) has increased compared to the pre—COVID level [21]. Before the COVID-19 lockdown, the proportion of sellers selling via phone call was 2.20% which more than doubled to reach 4.55% in post-COVID–19. The weekly average

**Table 1. Covid-19 impact on business modality.**

| Variables | | Pre-COVID (February, 2020) | During COVID-29 lockdown | Post Covid-19, recent 7 Days (August, 2021) |
|---|---|---|---|---|
| Number of people Engaged in the Business | Average Number of Working People | 1.4 | - | 1.33 |
| Online Business | Online Business (% of Yes) | 2.20% | - | 4.55% |
| Weekly Business Duration (in Number of Days) | Average Number of Days | 6.67 | 6.19 | 6.8 |
| Daily Business Duration (in Hours) | Average Number of Hours | 12.38 | 8.46 | 12.36 |
| | Median Number of Hours | 14 | 8 | 13 |
| Daily Average Sale (In BDT) | Average Daily Sale (BDT) | 10655 | 6224 | 7444 |
| | Median Daily Sale (BDT) | 9000 | 5000 | 6000 |
| Profit Information | Daily Average Profit Margin | 21% | 17% | 19% |

number of days and daily business duration in hours were asked at three points in time; pre-COVID–19 period, during the COVID–19 period, and the week before the survey. On average, they were operating weekly for 6.67 days before the COVID–19, which was reduced to 6.18 days during COVID. The reduction of weekly average days of business operation during the COVID period was estimated to be significant (two sample proportion test; $p < 0.05$). Also, 92% of the retailers were operated 7 days a week in pre-COVID time, which was reduced to only 56% in the COVID period. In terms of daily hours of business operation, in normal time the average was 12.38 hours and that was reduced to 8.46 hours during pandemic. The 42% average reduction in daily sales during the pandemic registered average sale to reach at BDT 6,224 from the pre-COVID-19 average level of BDT 10,655. This difference in sales was also statistically significant (two sample mean test, $p < 0.05$). The daily average profit margin, which was 21% in normal time reduced to 17% during crisis and this 2 percentage points reduction was also significant (two sample mean test, $p < 0.05$).

Table 2 displays the retailers' perception of the impact of COVID-19 on their sales and profit margin, and they were also asked about the reasons whether it was due to pandemic or seasonal or other causes. Regardless of the product they sale, around 87% of the retailers claimed that their sales were being affected during the COVID-19 pandemic. Among those sellers, 95% said that pandemic is the main reason for the change in sales, and the second main reason was the change in income of buyers (71% of the vegetable retailers and 68% of the fruit retailers). Asking about the perceived magnitude of changes in sales, 58% of vegetable retailers said that their sales reduced moderately (10% to 30%), and 59% of fruit retailers also mentioned similar reduction.

In case of the changes in profit margin, 58% of vegetable retailers said that their profit margin has changed during the COVID—19 period, while a similar response came from 70% of

**Table 2. Perceived impact of COVID-19 on sales and profit margin.**

| Effects | | Vegetable retailers (%) (n = 864) | Fruits retailers (%) (n = 468) |
|---|---|---|---|
| Any Change in Sales During COVID | Yes | 87.04 | 86.75 |
| | No | 12.96 | 13.25 |
| Magnitude of Change in Sales | Increased Highly (>30%) | 0.12 | 1.5 |
| | Increased Moderately (10% to 30%) | 2.43 | 2.99 |
| | Same as before | 12.96 | 13.25 |
| | Reduced moderately (10% to 30%) | 57.99 | 58.55 |
| | Reduced highly (>30%) | 25.81 | 22.86 |
| | Not Applicable | 0.69 | 0.85 |
| Reasons of Change in Sales | Seasonal | 16.2 | 18.8 |
| | Pandemic | 95.49 | 95.09 |
| | Change in Income | 70.83 | 68.59 |
| | Change in Price | 42.82 | 31.84 |
| | Change in Supply | 19.33 | 11.97 |
| | Others | 1.62 | 0.85 |
| Any Change in Profit Margin During COVID | Yes | 57.75 | 69.87 |
| | No | 42.25 | 30.13 |
| Magnitude of Change in Profit Margin | Increased Highly (>30%) | 0.23 | 1.5 |
| | Increased Moderately (10% to 30%) | 5.21 | 5.13 |
| | Same as before | 42.25 | 30.13 |
| | Reduced moderately (10% to 30%) | 42.94 | 52.14 |
| | Reduced highly (>30%) | 8.8 | 10.9 |
| | Not Applicable | 0.58 | 0.21 |

the retailers of fruit. Those who experienced change in profit margin, among them 43% of retailers of vegetable mentioned about moderate change (10% to 30%), while such moderate change was felt by 52% of the fruit retailers.

### 3.3 Regression analysis

The logistic regression estimation results in Table 3 show that the business location of the retailer is an important variable for understanding the level of disruption by COVID–19. It is due to the difference in strictness of the lockdown between the mega cities and medium-sized cities. The strict law enforcement during the lockdown period in Dhaka significantly affected the vegetable and fruit retailers compared to their counterparts in Manikganj. According to the findings, the likelihood of facing a reduction in the profit margin is lower for a retailer located in Manikganj (a medium-sized city) compared to one located in Dhaka (a mega city).

The probability of facing a reduction in profit margin was higher for fruit retailers than the retailers of vegetables, whereas the likelihood was lower for a retailer who sells both. The result is plausible since vegetables are necessary commodities whereas fruits are considered luxury

**Table 3. Impact estimation of COVID restriction on the vegetables and fruits retailers.**

| Independent Variables | | Marginal Effects from Logistic Regression (Dependent Variable: Reduction in Profit Margin) | | Ordinary Least Square (OLS) Regression (%age Change in Sales in BDT) | | | |
|---|---|---|---|---|---|---|---|
| | | | | Model 1 | | Model 2 | |
| | | Coeff. | P–Value | Coeff. | P–Value | Coeff. | P–Value |
| Item Type (*Ref: Only Vegetables*) | Only Fruits | 0.052* | 0.065 | -0.455 | 0.713 | -0.552 | 0.653 |
| | Vegetables, Fruits and Others | -0.177* | 0.091 | 3.005 | 0.580 | 2.563 | 0.635 |
| Retailer Type (*Ref: Street or Mobile Vendors*) | Wet Market | -0.017 | 0.547 | 1.533 | 0.222 | 1.495 | 0.231 |
| Business Location (*Ref: Dhaka*) | Manikganj | -0.278*** | 0.000 | 19.962*** | 0.000 | 19.254*** | 0.000 |
| Licence (*Ref: Available*) | Not Available | 0.041 | 0.463 | 4.839** | 0.037 | 4.357* | 0.059 |
| Educational Attainment (*Ref: No Formal Education*) | Primary | 0.169*** | 0.000 | -2.344* | 0.074 | -2.255* | 0.085 |
| | SSC/HSC/Equivalent | 0.167*** | 0.000 | -3.011* | 0.063 | -2.321 | 0.152 |
| | Degree/Hons'/Above | 0.175 | 0.194 | -2.466 | 0.692 | -2.824 | 0.648 |
| Gender (*Ref: Female*) | Male | 0.202** | 0.021 | 1.254 | 0.746 | 1.625 | 0.674 |
| Online Business (*Ref: Not Available*) | Available | -0.039 | 0.561 | 2.268 | 0.400 | 2.328 | 0.385 |
| Business Experience in Years | | 0.005 | 0.267 | -0.086 | 0.314 | -0.582*** | 0.009 |
| Number of Workers Available | | 0.127*** | 0.035 | -1.326 | 0.122 | -1.099 | 0.199 |
| Age of the Retailer in Years | | -0.013*** | 0.000 | -0.259*** | 0.000 | 0.671** | 0.011 |
| Square of the Retailer's Age | | - | - | - | - | -0.011*** | 0.000 |
| Square of the Retailer's Business Experience | | - | - | - | - | 0.016** | 0.016 |
| Constant | | - | - | -49.73*** | 0.000 | -65.10*** | 0.000 |
| Regression Diagnosis | Observation | 1277 | - | 1238 | - | 1238 | - |
| | Pseudo/R–Square | 0.123 | - | 0.147 | - | 0.158 | - |
| | LR Chi–Square | 215.49*** | 0.000 | - | - | - | - |
| | Area Under ROC | 0.716 | - | - | - | - | - |
| | F–Stat | - | - | 16.32*** | 0.000 | 15.30*** | 0.000 |
| | Average VIF | - | - | 1.21 | - | 6.15 | - |
| | Heteroscedasticity (Chi–Square) | - | - | 2.93* | 0.086 | 1.48 | 0.224 |
| | RESET (F–Stat) | - | - | 6.53*** | 0.000 | 2.30* | 0.076 |

Note

***, ** and * indicates 1%, 5% and 10% level of significance.

goods for many people, especially during the lockdown period when most people lost a significant amount of their income and faced the uncertainty of employment [22]. The likelihood of facing a reduction in profit margin was higher for the retailers with primary and SSC/HSC/Equivalent level of education compared to the seller having no formal education. The age of the business owner also had a significant and negative relationship with the dependent variable, indicating that the likelihood of facing a reduction in profit margin was lower for the higher—aged retailers. The number of workers employed in the business was positively related to the dependent variable, meaning the likelihood of facing a reduction in profit margin was increasing with the increase in the number of workers.

The estimation results provide evidence of the statistical significance of the business location of vegetables and fruit retailers on their percentage change in sales. More specifically, holding all other factors constant, the retailers located in Manikganj faced on average 19 percentage points less reduction in their sales in comparison to their counterparts doing business in Dhaka. Implying that retailers in Dhaka were more severely affected due to the COVID restrictions. The license variable is statistically significant at the 10% level, indicating that the percentage change in the reduction of sales was lower by 4 percentage points for the retailers with a business license than those without the business license. The reduction in sales is 2.55 percentage points higher for the retailers with the education level of primary compared to the seller without having any formal education. It could be due to the fact that educated retailers remain more conscious about COVID and reduce their business operation duration resulting in a fall in sales. Similar estimation results for the age variable are also observed, with lower sales for the higher-aged retailers.

In terms of regression diagnosis, both the logistic regression and OLS regression results in Model 2 are valid. The value for logistic regression and *F–Statistic* value for OLS regression in Model 2 is statistically significant establishing the overall significance of the estimation results. The data fitness of the logistic regression is established since the area under ROC is 0.716, higher than 0.50 (S1 Fig). In OLS regressions both in Model 1 and Model 2 the extent of multicollinearity is tolerable as the average VIF is less than 10 [17]. Model 1 has the problem of heteroscedasticity as the value of Breusch-Pagan $\chi^2$ is statistically significant, however, that problem is removed in Model 2. As Ramsey's RESET *F-Statistic* in Model 1 is statistically significant at 1% level, it has a specification problem with omitted variable bias. This particular problem is minimized in Model 2 as the Ramsey's RESET *F–Statistic* magnitude is reduced to a level making it non-rejectable at 5% level. Hence, the robustness of OLS regression results under Model 2 is more than in Model 1.

## 4. Discussion

The diverse nature of vegetables and fruits, and their varying supply chain often restrict implementing the fiscal and regulatory measures. In Bangladesh, vegetables and fruits are generally sold at traditional WM, though SUP are becoming popular in recent years with increased urbanization. Also, in Bangladesh, within fruits and vegetables supply chain, many actors plays crucial role including several intermediaries like locally called *bepari*, *aratdar* and wholesaler before reaching the retailers [23]. The COVID–19 restrictions in Bangladesh have adversely affected the supply chain [23], including one of the most vital players of the chain i.e. "retailers". Since the recommended amount of consumption of fresh fruits and vegetables by low-income people are relatively lower than the higher income group [24, 25], such disruption in the supply chain and shock on the active players might increase the price and hence further degrade the accessibility of healthy foods to the city dwellers. This study sought to investigate the impact of COVID-19 restrictions on the business and profitability of retailers of fruits and

vegetables and how government support is necessary for the retailers to absorb the shock, and consequently control the disruption in the supply chain and price of fruits and vegetables.

Findings revealed that due to restrictions, the retailers of fruits and vegetables faced reduction in sales (around 42%) and profit loss like other small and medium entrepreneurs in Bangladesh [26]. The COVID–19 agricultural stimulus packages ignored the financial support for the vegetables and fruits retailers. In the absence of government support, they become more vulnerable as there is a lack of sufficient resources, financial capacities, and shock resilience. Their income is also highly dependent on their daily sales from a small number of consumers [27]. So, their business survival capability due to the unprecedented COVID-19 pandemic is very low compared to larger business enterprises. Street or mobile vendors don't have enough financial and business infrastructural and access to the formal capital market compared to the well-established grocery store that sells similar types of products along with other products [28]. Impact of COVID-19 on the sales of fruits and vegetables is associated with the compromised livelihood of retailers [7].

This study also found that the fruit retailers have higher chance of facing reduction in profit margin than the retailers of vegetables since vegetables are comparatively more necessary goods than fruits. Moreover, this study showed that, due to COVID-19 restrictions, the fruits and vegetables retailers in large city (like Dhaka) have higher reduction in sales and profit loss compared to retailers in small city (like Manikganj). These findings might be crucial for developing government support program for the retailers and others small and medium business owners.

Developing strategies to promote accessibility and availability of fruit and vegetables round the year for all people, and subsequent utilization by women, children, poor and distressed households have been identified as a priority in Bangladesh [4, 29]. Instability in markets resulting in food-price spikes often influences access to fresh food, especially by the poor and disadvantaged [2]. A higher price means lower consumption of healthy items and a higher number of food-insecure households [30–32]. Government of Bangladesh (GoB) has emphasized the importance of public procurement of food grains, price supports after harvest, public stock for distribution, transportation facilities, and production and market monitoring along with enforcement of regulatory framework for reducing volatility of food price and ensuring spatial and seasonal availability of food [2]. Along with such conventional initiatives from the government, support mechanism targeting specific supply chain actors such as retailers of vegetables and fruits is imperative considering the COVID-19 context. Supply-side financial intervention is important to reduce vulnerability and ensure a smooth supply resulting in easy accessibility of vegetables and fruits.

The modelling exercise registered the importance of location of doing business for dietary foods. The varying severity of impact due to COVID is an empirical contribution to the theoretical field of research concerning "location model" [33]. Also, the findings has valid policy implication from practical perspective elucidating that for fiscal intervention location consideration as well as the consideration of product type is of vital importance to make the realistic impact on the group of interest. The study design, implementation and the analysis has the rigor to establish the findings. Nevertheless, it has few limitations too. The sampling frame included only two urban areas from Bangladesh thus generalization of the findings is limited. The impact was examined for the retailers only, however such impact on the other supply chain actors and on the consumers are obvious. All these can be considered as further scope of research.

## 5. Conclusions

The current study aimed towards examining the impact of COVID–19 on the retailers of vegetables and fruits in the urban areas of Bangladesh. A primary research was designed to cross

sectionally survey 1,319 vegetables and fruits retailers in two densely populated urban regions in Bangladesh. Besides quantifying the presence and magnitude of COVID—19 impact on the vegetables and fruits retailers, the data analysis strived to answer whether such impact varied with regard to the location of business operation, type of business item and retailers type. While the country is experiencing an upward trend in the NCDs, any factors affecting the supply chain actors of healthy food is projected to degrade the situation. It was established that the location of the business (Dhaka or Manikganj) and type of business item (vegetables or fruits or both) are important while analyzing the reduction in profit margin and percentage change in sales. However, the type of retailer (WM or SS, or MV) is not significant. Thus, area-specific and product-specific interventions are required for the retailers of vegetables and fruits to minimize their vulnerability. Designing and implementing the such government interventions can result in price stability of dietary foods such as vegetables and fruits leading to the increase in their intake and help combating NCDs.

## Supporting information

**S1 Table. Mapping of vegetables and fruits POS retailers and markets in Dhaka and Manikganj.**
(DOCX)

**S2 Table. Socio-demographic and business profile of the vegetables and fruits retailers.**
(DOCX)

**S3 Table. Vegetables and fruits retailers dataset.**
(DTA)

**S1 Fig. Area under ROC curve for logistic regression.**
(TIF)

## Author Contributions

**Conceptualization:** Md. Nazmul Hossain, Md. Saiful Islam, S. M. Abdullah, Syed Mahbubul Alam, Rumana Huque.

**Data curation:** Md. Nazmul Hossain, Md. Saiful Islam, S. M. Abdullah.

**Formal analysis:** Md. Nazmul Hossain, Md. Saiful Islam, S. M. Abdullah.

**Funding acquisition:** Syed Mahbubul Alam, Rumana Huque.

**Investigation:** Md. Nazmul Hossain, Md. Saiful Islam, S. M. Abdullah, Rumana Huque.

**Methodology:** Md. Nazmul Hossain, Md. Saiful Islam, S. M. Abdullah, Rumana Huque.

**Project administration:** Syed Mahbubul Alam, Rumana Huque.

**Resources:** Rumana Huque.

**Software:** Md. Nazmul Hossain, Md. Saiful Islam, S. M. Abdullah.

**Supervision:** Md. Nazmul Hossain, S. M. Abdullah, Syed Mahbubul Alam, Rumana Huque.

**Validation:** Md. Nazmul Hossain, Md. Saiful Islam, S. M. Abdullah.

**Visualization:** Md. Nazmul Hossain, Md. Saiful Islam, S. M. Abdullah.

**Writing – original draft:** Md. Nazmul Hossain, Md. Saiful Islam, S. M. Abdullah, Rumana Huque.

**Writing – review & editing:** Md. Nazmul Hossain, Md. Saiful Islam, S. M. Abdullah, Syed Mahbubul Alam, Rumana Huque.

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
