## [Decision Letter · Decision Letter 0]

16 Aug 2022

PONE-D-22-20793Vegetables and fruits retailers in two urban areas of Bangladesh: Disruption due to COVID – 19 and implications for NCDsPLOS ONE

Dear Dr. Abdullah,

Thank you for submitting your manuscript to PLOS ONE. After careful consideration, we feel that it has merit but does not fully meet PLOS ONE’s publication criteria as it currently stands. Therefore, we invite you to submit a revised version of the manuscript that addresses the points raised during the review process.

We look forward to receiving your revised manuscript.

Kind regards,

Muhammad Mohiuddin

Academic Editor

PLOS ONE

Journal Requirements:

2. We note that Figure 1 in your submission contain [map/satellite] images which may be copyrighted. All PLOS content is published under the Creative Commons Attribution License (CC BY 4.0), which means that the manuscript, images, and Supporting Information files will be freely available online, and any third party is permitted to access, download, copy, distribute, and use these materials in any way, even commercially, with proper attribution. For these reasons, we cannot publish previously copyrighted maps or satellite images created using proprietary data, such as Google software (Google Maps, Street View, and Earth). For more information, see our copyright guidelines: http://journals.plos.org/plosone/s/licenses-and-copyright.

Additional Editor Comments:

Dear Authors,

W have received two assessment reviews of your paper. Though the reviewers comments were not that much critical but my own reading of the paper found that you need further improvement of the paper before we can proceed for publication. please consider the followings:

1. please reply in details of the issues raised by the two reviewers.

2. please highlight your Novelty of this study. Why do you consider that this paper is bringing something new that we do not know yet? how your findings or methodologies are innovative and contribute to the current knowledge?

3. please improve the discussion section by adding comparison and contrasting with current literature.

4. please explain clealarly both theoretical and practical contributions.

5. please Edit the paper before re-submision.

Reviewers' comments:

Reviewer's Responses to Questions

**Comments to the Author**

1. Is the manuscript technically sound, and do the data support the conclusions?

Reviewer #1: Yes

Reviewer #2: Yes

2. Has the statistical analysis been performed appropriately and rigorously? 

Reviewer #1: Yes

Reviewer #2: Yes

3. Have the authors made all data underlying the findings in their manuscript fully available?

Reviewer #1: Yes

Reviewer #2: Yes

4. Is the manuscript presented in an intelligible fashion and written in standard English?

Reviewer #1: Yes

Reviewer #2: Yes

5. Review Comments to the Author

Reviewer #1: Several comments have been made regarding this paper that need to be improved. Appropriate numbering for each section need to be considered if needed

a) Introduction

i) The introduction section is shallow. An extended discussion of the topic of the study, and the necessary aspects and keywords need to be discussed at length to set a comprehensive background of the study

ii) The paper has no objectives. You need to mention and discuss the objectives of the study from the definition of the research problem

b) Literature Review

iii) I also advice that you need to include the literature review, and discuss the findings of the previous studies, and the research methodologies applied

c) Material and Methods

i) This section is well presented and discussed, in terms of study design, study site and sampling frame, Sampling Strategy, Survey Participants, and Sample Size as well as Field Design and Implementation, Data Collection Tools, and Data management

ii) The data analysis techniques adopted in the study is also discussed

iii) However, this section takes a big section as compared to the previous section. I believe that this section can be streamlined to flow and add the other previous ‘introduction’ and ‘literature review’ section

d) Results

i) The result section is well presented

ii) However, I believe that the section need to be numbered. Numbering like 4.1, 4.2 etc to establish a flow of subsections

iii) In writing the percentage in symbols, the numbers and percentage symbol should be one word not two words. i.e

In line 252 it should be 2.20% and not ‘2.20 %’

e) Discussion

i) The discussion of the study is well presented in terms of addressing the topic of the study; however, please consider citing the following studies:

Hoque, A., Mohiuddin, M., & Su, Z. (2018). Effects of Industrial Operations on Socio-Environmental and Public Health Degradation: Evidence from a Least Developing Country (LDC). Sustainability, 10(11), 3948. https://doi.org/10.3390/su10113948

Chaveesuk, S., Khalid, B., & Chaiyasoonthorn, W. (2022). Continuance intention to use digital payments in mitigating the spread of COVID-19 virus. International Journal of Data and Network Science, 6(2), 527–536. https://doi.org/10.5267/j.ijdns.2021.12.001

Khalid, B., & Kot, M. (2021). The Impact of Accounting Information Systems on Performance Management in the Banking Sector. IBIMA Business Review, 1–15. https://doi.org/10.5171/2021.578902

ii) The previous literature findings has also been addressed in relation to the findings of the current study

f) Conclusions

i) The conclusions of the study does not seem to address the relevant information

ii) The conclusion should summarize the findings of the study. it should also state whether the set objectives of the study has been met or not

Two section should also be included in addition to conclusion

i) The study implications discussing both managerial implications and theoretical implications

ii) The recommendations for future studies

Reviewer #2: A study of 1,319 retailers was undertaken in two urban regions, namely Dhaka and Manikganj, between September 2021 and October 2021 with the goal of determining the influence of COVID-19 on the business practices and outcomes for the vegetables and fruit retailers in Bangladesh. This manuscript has met the quality requirements for publication in 'PLoS ONE' I wish the writers continued success in their efforts to improve the paper.

6. PLOS authors have the option to publish the peer review history of their article (what does this mean?). If published, this will include your full peer review and any attached files.

Reviewer #1: No

Reviewer #2: **Yes: **Mohammad Delwar Hussain

Green Business School

Green University of Bangladesh

---

## [Author Response · Author response to Decision Letter 0]

30 Sep 2022

Vegetables and fruits retailers in two urban areas of Bangladesh: Disruption due to COVID – 19 and implications for NCDs

Dear Respected Editor,

On behalf of all the authors, I thank the academic editor, all the reviewers and the editorial team for their comments and feedback on our manuscript. We feel that the reviewer’s comments have been really helpful in rectifying the gaps in the manuscript. We have tried our best to address each comment and incorporate the suggested changes in the revised manuscript. Our responses to the academic editor, reviewers and the editorial team and the corresponding edits can be found below: 

Response to the Editorial Team on Journal Requirements

Observation 1: Please ensure that your manuscript meets PLOS ONE's style requirements, including those for file naming. 

Response to Observation 1:

The revised manuscript followed the PLOS ONE style templates for formatting the Title, Authors Affiliation and Main body of the text. We believe the format is now in line with the PLOS ONE journal style and policy.

Observation 2: We note that Figure 1 in your submission contain [map/satellite] images which may be copyrighted. All PLOS content is published under the Creative Commons Attribution License (CC BY 4.0), which means that the manuscript, images, and Supporting Information files will be freely available online, and any third party is permitted to access, download, copy, distribute, and use these materials in any way, even commercially, with proper attribution. For these reasons, we cannot publish previously copyrighted maps or satellite images created using proprietary data, such as Google software (Google Maps, Street View, and Earth). We require you to either (1) present written permission from the copyright holder to publish these figures specifically under the CC BY 4.0 license, or (2) remove the figures from your submission:

a) You may seek permission from the original copyright holder of Figure 1 to publish the content specifically under the CC BY 4.0 license.

b) If you are unable to obtain permission from the original copyright holder to publish these figures under the CC BY 4.0 license or if the copyright holder’s requirements are incompatible with the CC BY 4.0 license, please either

i) remove the figure or 

ii) supply a replacement figure that complies with the CC BY 4.0 license.

Response to Observation 2: 

Thanks to the editorial team for raising this issue and guiding us to solve it accordingly. The Figure 1 is now replaced with a new version of graph drawn by the authors which is incompliant with CC BY 4.0 license. The drawing was done by authors themselves using required district wise and subdistrict wise shape file of Bangladesh (available in https://data.humdata.org/dataset/cod-ab-bgd). Also authors utilised Python software for drawing the maps. It can now be considered as property of this research and has no legal bindings for publication. The figure is replaced in PP7 in Line 139 under section named “Study Sites and Sampling Frame”.

Response to the Editorial Additional Comments 

Comment 1: Please reply in details of the issues raised by the two reviewers. 

Response to Comment 1:

The reviewer comments have been addressed accordingly in the revised manuscript. We would like to refer to the “Response to the Reviewer Comments” of this rebuttal letter as an endorsement.

Comment 2: Please highlight your Novelty of this study. Why do you consider that this paper is bringing something new that we do not know yet? How your findings or methodologies are innovative and contribute to the current knowledge? 

Response to Comment 2: 

Thanks for raising a very valid point regarding research. As it has been discussed in the Introduction section of the paper, considering the uprising of NCD problem in Bangladesh the demand side and as well as supply side intervention for promoting the dietary foods is unregistered. In order to design the proper and effective fiscal and public health intervention it is imperative to know the implementation challenges and the practicalities of the actors. To the best of our knowledge the current research is the only novel effort that tried to study the most important supply chain actor of fruits and vegetables in urban areas of Bangladesh with rigor and extent. Also, we knew that the prices of fruits and vegetables were affected due to the COVID-19 from existing research but how the event affected the retailers in Bangladesh and whether the affect remained same everywhere or not is unknown. Most of the research in this arena is directed toward identifying the implication for food security instead of searching for the impact on retailers. The approach followed is innovative as its inclusive of all different types of retailers for fruits and vegetables in Bangladesh and for measuring the impact besides considering all the conventional variables it also included the “location aspect”, “Retailer Type Aspect” and the “Type of Diet i.e. Fruits vs. vegetable aspect”. Also, to meet the research objectives the exercise answers definite research questions that registered the rigor. These facts are mentioned exclusively in the revised version of manuscript under the section introduction. For instance:

In PP3 Line 57-61 it mentions,

“Food stamp program or supplementary nutrition program is widely applied in developed countries to make fruits and vegetables affordable for low-income households. However, a healthy diet incentive program is not much prominent in underdeveloped and developing countries and Bangladesh is not an exception. Additionally, the supply-side intervention to enhance the uptake of such diet is even scarce”

In PP3-4 Line 64-67 it mentions,

“The imposition of lockdown and maintenance of social distancing created a disturbance …………Both the demand and supply side of the fruits and vegetable industry are disrupted due to the COVID-19 pandemic, but the direction and magnitude of the impact are ambiguous.[7]”

In PP4 Line 80-82 it mentions,

“Considering the increasing prevalence of NCDs, little is known about the supply chain actors of healthy foods such as vegetables and fruits in urban areas and the impact of fiscal policies and regulatory measures in Bangladesh.”

In PP4-5 Line 85-87 and 95-97 it mentions

“There is lack of evidence in Bangladesh on the impact of COVID-19 on the fruits and vegetables retailers in terms of the magnitude of profit margin, daily sales, and business operation……. This paper contributes to the current literature in terms of examining the impact of COVID-19 on the fresh fruits and vegetables retailers in the urban area of Bangladesh.”

In PP5 Line 98-101 it mentions

“……..the research questions as follows were the fruits and vegetables retailers being affected by COVID-19?, did the impact vary with retailers characteristics? and what was the magnitude of impact compared to the pre-COVID level in terms of daily sales, profit margin, business operation, etc?”

Comment 3: Please improve the discussion section by adding comparison and contrasting with current literature. 

Response to Comment 3: 

Thanks for the guidance provided. We followed the instruction also in accordance with suggestion made by Reviewer 1. The revised version of the manuscript included an updated discussion part where the comparison and contrast performed where seems required and also the arguments were defended using the existing research evidences. For instance, 

In Discussion PP19 Line 396 – 398 it mentions using existing research evidence:

“Findings revealed that due to restrictions, the retailers of fruits and vegetables faced reduction in sales (around 42%) and profit loss like other small and medium entrepreneurs in Bangladesh [4, 29]”

In PP18 Line 377 – 379 it mentions using existing research evidence:

“Developing strategies to promote accessibility and availability of fruit and vegetables round the year for all people, and subsequent utilization by women, children, poor and distressed households have been identified as a priority in Bangladesh.[26]”

In PP16 Line 366 – 368 it mentions using existing research evidence:

“…Also, in Bangladesh, within fruits and vegetables supply chain, many actors plays crucial role including several intermediaries like locally called bepari, aratdar and wholesaler before reaching the retailers…[23]”

In PP18-19 Line 380 – 388 it mentions using existing research evidence:

“In the absence of government support, they become more vulnerable as there is a lack of sufficient resources, financial capacities, and shock resilience. Their income is also highly dependent on their daily sales from a small number of consumers[27]. So, their business survival capability due to the unprecedented COVID-19 pandemic is very low compared to larger business enterprises. Street or mobile vendors don’t have enough financial and business infrastructural and access to the formal capital market compared to the well-established grocery store that sells similar types of products along with other products[28]. Impact of COVID-19 on the sales of fruits and vegetables is associated with the compromised livelihood of retailers[7].”

Comment 4: Please explain clealarly both theoretical and practical contributions. 

Response to Comment 4: 

The discussion part of the revised manuscript contains the suggested amendment. Authors included the empirical contribution to the intended theoretical field and also provided the practical and policy implications. It also mentioned about the strength of the research along with potential limitation and future scope of research. 

In discussion section PP20 Line 410 – 420 includes the following to meet the purpose:

“The modelling exercise registered the importance of location of doing business for dietary foods. The varying severity of impact due to COVID is an empirical contribution to the theoretical field of research concerning “location model”[33]. Also, the findings has valid policy implication from practical perspective elucidating that for fiscal intervention location consideration as well as the consideration of product type is of vital importance to make the realistic impact on the group of interest. The study design, implementation and the analysis has the rigor to establish the findings. Nevertheless, it has few limitations too. The sampling frame included only two urban areas from Bangladesh thus generalization of the findings is limited. The impact was examined for the retailers only, however such impact on the other supply chain actors and on the consumers are obvious. All these can be considered as further scope of research.”

Comment 5: Please Edit the paper before re-submision.

Response to Comment 5: 

All the formatting edits suggested by the editorial team has been performed following the PLOS ONE guideline in the revised manuscript. Regarding the edits in terms of content and clarifications all the necessities were addressed following the suggestion from reviewers and academic editor. All the changes are highlighted in green in the marked manuscript. The authors believe that the revised version has a substantial improvement as demanded by the reviewers and academic editors. 

Response to the Reviewer Comments

Comments from Reviewer 1:

Several comments have been made regarding this paper that need to be improved. Appropriate numbering for each section need to be considered if needed.

Response:

Authors are thankful to the anonymous reviewer for the valuable suggestions. Following the suggestion numbering is introduced for each section and subsections in the revised manuscript. Also, as suggested in Introduction under PP5 Line 101-104 we have mentioned the article structure by numbering the section as follows:

“The structure of this paper contains introduction in section 1, methodology of the paper in section 2, descriptive and econometric results are discussed in section 3 and finally the discussion based on the results and conclusions are presented in section 4 and section 5 respectively.”

Comments Regarding Introduction and Literature Review

Comment 1: The introduction section is shallow. An extended discussion of the topic of the study, and the necessary aspects and keywords need to be discussed at length to set a comprehensive background of the study. 

Response to Comment 1: 

Thanks for raising the issue. We have revised the introduction section following the guideline mentioned. The revised version included the discussion of aspects related to the study title and coherence has been maintained while extending the story to set the background. It now includes the discussion on important key words as follows:

In PP3 Line 45 and subsequent lines mentioned about NCD status,

“It is estimated that about 580,000 deaths are caused by NCD annually, representing more than 67% of all deaths in Bangladesh, and unhealthy lifestyles and diets play a leading role in this epidemic [1].”

In PP3 Line 50 and subsequent lines mentioned about fruits and vegetables intake status,

“In Bangladesh, overall daily per capita consumption of fruits was 1.7 servings and of vegetables was 2.3 servings [4]. Considering the daily total requirement of 5 servings as the minimum recommended amount [5], 95.7% of people do not consume adequate amount of fruits or vegetables on an average day [4]”

In PP3 Line 62 and subsequent lines mentioned about COVID – 19 and its related implications 

“….Households reduced their consumption of necessary commodities due to the adverse impact of COVID-19 and the lower income households were affected mostly compared to those from the richest quintile [6]. The imposition of lockdown and maintenance of social distancing created a disturbance in fresh fruits and vegetable production and their retailing [7]. Both the demand and supply side of the fruits and vegetable industry are disrupted due to the COVID–19 pandemic….”

In PP4 Line 69 and subsequent lines mentioned about type of retailers for fruits and vegetables and COVID – 19 implications and disruptions for them: 

“Besides the Wet Market (WM) and Super Shops (SUP), vegetables and fruits in the cities of lower and lower-middle-income countries are mostly sold by Street Shops (SS) and Mobile Vendors (MV). Fresh fruits and vegetable vendors play an important role in providing these dietary foods at a lower price, especially to the low income and vulnerable populations[8]. A comparison between the pre and post pandemic time reveals that, of the identified vendors before COVID–19, 35% of the vendors were absent or closed business during the pandemic[7].”

In PP4 Line 82 and subsequent lines mentioned about the work approach and knowledge gap: 

“This research conducted a survey among the vegetables and fruits retailers in Bangladesh with a general objective to explore the barriers and facilitators for promoting vegetables and fruit intake and to identify the gap in required fiscal and regulatory policy support. There is a lack of evidence in Bangladesh on the impact of COVID-19 on the fruits and vegetable retailers in terms of the magnitude of profit margin, daily sales, and business operation.”

Comment 2: The paper has no objectives. You need to mention and discuss the objectives of the study from the definition of the research problem.

Response to Comment 2:

In the revised version of manuscript, the introduction section now clearly mentions the research gap and the objective of the research. Also, it specifies the research questions addressing those which helped to meet the objective. 

In PP3-4 Line 64 and subsequent the research gap is mentioned as follows,

“The imposition of lockdown and maintenance of social distancing created a disturbance in fresh fruits and vegetable production and their retailing. Both the demand and supply side of the fruits and vegetable industry are disrupted due to the COVID-19 pandemic, but the direction and magnitude of the impact are ambiguous…… There is a lack of evidence in Bangladesh on the impact of COVID-19 on the fruits and vegetable retailers in terms of the magnitude of profit margin, daily sales, and business operation.”

In PP4-5 Line 87 and subsequent the research objective is mentioned as follows,

“…the current analysis aimed towards scrutinizing the impact of COVID-19 on the business practices and outcomes of vegetables and fruits retailers in Bangladesh……The existing pieces of literature mainly focused on the COVID-19 disruption and the corresponding implication on food security. This paper contributes to the current literature in terms of examining the direct impact of COVID-19 disruption on the fresh fruits and vegetables retailers in the urban areas of Bangladesh”

In PP5 Line 98 and subsequent the research questions are mentioned as follows,

“…were the fruits and vegetables retailers being affected by COVID-19?, did the impact vary with retailers characteristics? and what was the magnitude of impact compared to the pre-COVID level in terms of daily sales, profit margin, business operation, etc”

Comment 3: I also advice that you need to include the literature review, and discuss the findings of the previous studies, and the research methodologies applied. 

Response to Comment 3: 

Thanks for the raised concern. As per our understanding and what we have seen in the manuscript guideline for PLOS ONE the convention of including a separate section on “Literature Review” is not there. Also, several public health journals and recent articles did not have a specific “Literature Review” section. However, following the suggestion we built on our arguments throughout the paper with existing research evidences. Comparison and contrast with the current evidences has been performed while developing the discussion. Also, for the methodological development and argument established or doing proposition we have used the evidence of relevant updated literatures.

Comments Regarding Materials and Methods

Comment 1: This section is well presented and discussed, in terms of study design, study site and sampling frame, Sampling Strategy, Survey Participants, and Sample Size as well as Field Design and Implementation, Data Collection Tools, and Data management.

Response to Comment 1: 

Gratitude for the positive notes. The research team is glad knowing that the reviewer found the approach useful and depth is well regarded.

Comment 2: The data analysis techniques adopted in the study is also discussed.

Response to Comment 2: 

Thanks for the positive reflection. We are glad that the analysis technique is found to be complete and discussed properly.

Comment 3: However, this section takes a big section as compared to the previous section. I believe that this section can be streamlined to flow and add the other previous ‘introduction’ and ‘literature review’ section.

Response to Comment 3: 

Thanks for the observation. In the original manuscript this materials and methods section was relatively larger. Nevertheless, in the revised version, following the comments of academic editor and reviewers, as the introduction section is revised to improve the coherence and rationality by weaving from the review of existing literature, the materials and method section is in accordance with the earlier section in terms of depth and length. As mentioned in Comment 1 and Comment 2 by the reviewers because of being detail the readability of the methods and understanding was clear. This depth and detail are required also for the future research if anybody interested to replicate the work in other parts of the world. The authors are glad to mention that the study approach in this regard was itself quiet of a challenge in terms of implementation, thereby documenting it in proper detail will add into the filed too. 

Comments Regarding Results

Comment 1: The result section is well presented.

Response to Comment 1:

Thanks for the positive note. It’s encouraging for the authors that reviewers found the result section succinct and proper. 

Comment 2: However, I believe that the section need to be numbered. Numbering like 4.1, 4.2 etc to establish a flow of subsections.

Response to Comment 2:

Thank you for raising the issue. In the revised manuscript it has been addressed by introducing section and sub section number. Additionally, to make it distinct, the section main headings were made separated from the sub headings using a different font size using the PLOS ONE guideline. PLOS ONE suggested Level 1 section heading with 18 font and Level 2 section subheading with 16 font. 

Comment 3: In writing the percentage in symbols, the numbers and percentage symbol should be one word not two words. i.e in line 252 it should be 2.20% and not ‘2.20 %’.

Response to Comment 3:

Thanks for pointing out these typos. We have made all the necessary changes throughout the manuscript.

Comments Regarding Discussion

Comment 1: The discussion of the study is well presented in terms of addressing the topic of the study; however, please consider citing the following studies:

i) Hoque, A., Mohiuddin, M., & Su, Z. (2018). Effects of Industrial Operations on Socio-Environmental and Public Health Degradation: Evidence from a Least Developing Country (LDC). Sustainability, 10(11), 3948. https://doi.org/10.3390/su10113948

ii) Chaveesuk, S., Khalid, B., & Chaiyasoonthorn, W. (2022). Continuance intention to use digital payments in mitigating the spread of COVID-19 virus. International Journal of Data and Network Science, 6(2), 527–536. https://doi.org/10.5267/j.ijdns.2021.12.001

iii) Khalid, B., & Kot, M. (2021). The Impact of Accounting Information Systems on Performance Management in the Banking Sector. IBIMA Business Review, 1–15. https://doi.org/10.5171/2021.578902

Response to Comment 1:

The authors glad to know that the organization and development of the discussion section was found appropriate. Also, thanks for introducing the authors with some recent and relevant literatures. The suggested citations were made in accordance with their relevance. 

The papers are cited in the following statement under in PP13, Line 272 – 274 and in PP19 Line 396 – 398,

“The proportion of retailers selling fruits and vegetables online (via phone call) has increased compared to the pre - COVID level [21]”

“Developing strategies to promote accessibility and availability of fruit and vegetables round the year for all people, and subsequent utilization by women, children, poor and distressed households have been identified as a priority in Bangladesh[4,29]”

Comment 2: The previous literature findings has also been addressed in relation to the findings of the current study.

Response to Comment 2:

Thanks, and authors felt encouraged to know that reviewer found that the current findings are in line with the existing evidence. The findings of the existing literatures were referred to in due relevance. 

Comments Regarding Conclusion

Comment 1: The conclusions of the study does not seem to address the relevant information.

Response to Comment 1:

The revised version of the manuscript contains an updated version of the conclusion that addressed this observation. The conclusion is now properly summarized the objective and approach of the findings. It also summarized the findings and its relevant implications. The following amendments can be found in the new version of conclusion which reads as follows:

In PP20 - 21Line 422 – 436,

“The current study aimed towards examining the impact of COVID–19 on the retailers of vegetables and fruits in the urban areas of Bangladesh. A primary research was designed to cross sectionally survey 1,319 vegetables and fruits retailers in two densely populated urban regions in Bangladesh. Besides quantifying the presence and magnitude of COVID - 19 impact on the vegetables and fruits retailers, the data analysis strived to answer whether such impact varied with regard to the location of business operation, type of business item and retailers type. While the country is experiencing an upward trend in the NCDs, any factors affecting the supply chain actors of healthy food is projected to degrade the situation. It was established that the location of the business (Dhaka or Manikganj) and type of business item (vegetables or fruits or both) are important while analyzing the reduction in profit margin and percentage change in sales. However, the type of retailer (WM or SS, or MV) is not significant. Thus, area-specific and product-specific interventions are required for the retailers of vegetables and fruits to minimize their vulnerability. Designing and implementing the such government interventions can result in price stability of dietary foods such as vegetables and fruits leading to the increase in their intake and help combating NCDs.”

Comment 2: The conclusion should summarize the findings of the study. It should also state whether the set objectives of the study has been met or not. 

Two section should also be included in addition to conclusion:

i) The study implications discussing both managerial implications and theoretical implications

ii) The recommendations for future studies

Response to Comment 2:

The revised conclusion summarized the findings and relates it to the study objectives, which is self-explanatory in terms of accomplishing. Regarding the study implications along with strengths and limitations of the study which lead to the future scope of research has also been added. This addition has been included at the end of the discussion as a convention. It reads as follows:

In PP20 Line 410 – 420,

“The modelling exercise registered the importance of location of doing business for dietary foods. The varying severity of impact due to COVID is an empirical contribution to the theoretical field of research concerning “location model”[33]. Also, the findings has valid policy implication from practical perspective elucidating that for fiscal intervention location consideration as well as the consideration of product type is of vital importance to make the realistic impact on the group of interest. The study design, implementation and the analysis has the rigor to establish the findings. Nevertheless, it has few limitations too. The sampling frame included only two urban areas from Bangladesh thus generalization of the findings is limited. The impact was examined for the retailers only, however such impact on the other supply chain actors and on the consumers are obvious. All these can be considered as further scope of research.”

Comments from Reviewer 2:

A study of 1,319 retailers was undertaken in two urban regions, namely Dhaka and Manikganj, between September 2021 and October 2021 with the goal of determining the influence of COVID-19 on the business practices and outcomes for the vegetables and fruit retailers in Bangladesh. This manuscript has met the quality requirements for publication in 'PLoS ONE' I wish the writers continued success in their efforts to improve the paper.

Response to Reviewer 2:

Thanks, and gratitude to Reviewer 2 for the positive notes. The authors are encouraged as the reviewer have found that the manuscript has merit and quality for the Journal. 

Comments on Title: Impact of COVID-19, the second part of the paper's title, is preferable to having it in just one section.

Response to Comments on Title:

The authors appreciate the observation. Nevertheless, as the manuscript reads the research context is rooted with diet related Non-Communicable Diseases (NCDs) for Bangladesh. It has a great derived implication in the context of COVID-19 and the dwindling impact it had on the dietary supply chain. From that aspect authors choose that title which is to some extent self-explanatory. The authors would like to humbly request the anonymous reviewer to approve the existing title considering the ground of better understanding of the research context.

Comments on Abstract: The abstract needs to follow the summary of the Background, Methods, Findings, and Interpretation.

Response to Comments on Abstract: 

The revised version of the manuscript contains the abstract that covered the suggested summaries of respective sections. 

In PP2 Line 20 – 27 contains the summary of Background as 

“Bangladesh is experiencing an increasing prevalence of diet-related non-communicable diseases (NCDs). Considering daily total requirement of 5 servings as minimum recommended amount, 95.7% of people do not consume adequate fruit or vegetables on an average day in the country. Imposition of lockdown during COVID-19 created disturbance in fresh fruits and vegetable production and their retailing. This incident can make these dietary products less affordable by stimulating price and trigger NCDs. However, little is known about the supply chain actors of healthy foods such as vegetables and fruits in urban areas, and how they were affected due to pandemic.”

In PP2 Line 27 – 32 contains the summary of Method as 

“Aiming toward the impact of COVID–19 on the business practices and outcomes for the vegetables and fruits retailers in Bangladesh, a survey of 1,319 retailers was conducted in two urban areas, namely Dhaka and Manikganj from September 2021 to October 2021. To comprehend the impact of COVID-19 on the profit margin of the retailers and on the percentage change in sales, a logistic and an Ordinary Least Squares (OLS) regression were estimated.”

In PP2 Line 32 – 40 contains the summary of Results as 

“Significant difference in the weekly business days and daily business operations was observed. The average daily sales were estimated to have a 42% reduction in comparison to pre-COVID level. The daily average profit margin on sales was reportedly reduced to 17% from an average level of 21% in the normal period. Nevertheless, this impact is estimated to be disproportionate to the product type and subject to business location. The probability of facing a reduction in profit margin is higher for the fruit sellers than the vegetable sellers. Contemplating the business location, the retailers in Manikganj (a small city) faced an average of 19 percentage points less reduction in their sales than those in Dhaka (a large city).”

In PP2 Line 40 – 42 contains the summary of conclusion and implication as 

“Area-specific and product-specific intervention are required for minimizing the vulnerability of retailers of vegetables and fruits and ensuring smooth supply of fruits and vegetables and increasing their uptake to combat diet related NCD.”

Comments of Introduction: The rationale for the study is explained in the introduction section, but the study's gap with respect to prior research, contribution to the global context, and guidance throughout the paper are all absent.

Response to Comments of Introduction:

Thanks for this important observation. The introduction section was revised accordingly to address the problem. As it now reads,

In PP4-5 Line 85 – 97 mentions the research gap and contribution 

“There is a lack of evidence in Bangladesh on the impact of COVID-19 on the fruits and vegetable retailers in terms of the magnitude of profit margin, daily sales, and business operation………..Functioning of the value chain of perishable goods were highly affected due to the imposition of countrywide lockdown[9,10].Because of this downturn many people involved in this sector lost their job not only in Bangladesh but also in other countries [11]. The research evidence revealed that there was a significant and negative impact of COVID-19 on the prices of fruits and vegetables [12]. The existing pieces of literature mainly focused on the COVID-19 disruption and the corresponding implication on food security. This paper contributes to the current literature in terms of examining the direct impact of COVID-19 disruption on the fresh fruits and vegetables retailers in the urban areas of Bangladesh.”

In PP5 Line 98 – 104 mentions the research questions and guidance of the paper

“The research objective was met answering the questions as follows, were the fruits and vegetables retailers being affected by COVID-19?, did the impact vary with retailers characteristics? and what was the magnitude of impact compared to the pre-COVID level in terms of daily sales, profit margin, business operation, etc? The structure of this paper contains introduction in section 1, methodology of the paper in section 2, descriptive and econometric results are discussed in section 3 and finally the discussion based on the results and conclusions are presented in section 4.”

Comments on Research Methodology: If the sample had been a bit stronger, the base would have been more robust. It might be able to uncover more precise explanations for the retailers, for instance, if certain rural and urban parts of the nation were compared and improved.

Response to Comments on Research Methodology:

The authors appreciate the observation and fully agree with the spirit. However, as primary research and since the NCDs problem is more prevalent in urban areas in Bangladesh the design was made to reveal the affect in that very geographic region because of time and fund constraint. In the revised version of the manuscript, we have admitted this as a limitation of the current research and as research implication suggested towards further future research in this aspect. 

In PP20 Line 415 – 420 it now includes

“The study design, implementation and the analysis has the rigor to establish the findings. Nevertheless, it has few limitations too. The sampling frame included only two urban areas from Bangladesh thus generalization of the findings is limited. The impact was examined for the retailers only, however such impact on the other supply chain actors and on the consumers are obvious. All these can be considered as further scope of research.” 

Comments on Analysis of Empirical Results: The results are presented in a proper and understandable way. The findings and the results are consistent.

Response to Comments on Analysis of Empirical Results:

Thanks for the positive note. Authors appreciate reviewer’s views and sincerely glad that the section was found understandable and consistent.

Comments on Discussion: The authors incorporate all of the discussion based on the empirical data, although it must be evaluated in light of earlier work.

Response to Comments on Discussion:

The revised version of the manuscript has a discussion which has presented the interpretation incorporating the relevant literatures to compare and contrast duly. However, it is worth mentioning that there is scarcity of literature in the design and approach followed to address the research problem and hence making any direct comparison remained as a remote possibility. Nevertheless, the authors gave effort and tried to address the observation considering the context of the findings and also focused on the implication. As few instances, in the discussion it now reads,

In PP13, Line 272 – 274 and in PP19 Line 396 – 398 findings were referred in accordance with existing research evidences:

“The proportion of retailers selling fruits and vegetables online (via phone call) has increased compared to the pre - COVID level [21]”

“Developing strategies to promote accessibility and availability of fruit and vegetables round the year for all people, and subsequent utilization by women, children, poor and distressed households have been identified as a priority in Bangladesh[4,29]”

In PP18, Line 377 – 379 similar approaches followed

“Findings revealed that due to restrictions, the retailers of fruits and vegetables faced reduction in sales (around 42%) and profit loss like other small and medium entrepreneurs in Bangladesh[26].”

In PP19, Line 389 – 395 similar approaches followed

“This study also found that the fruit retailers have higher chance of facing reduction in profit margin than the retailers of vegetables since vegetables are comparatively more necessary goods than fruits. Moreover, this study showed that, due to COVID-19 restrictions, the fruits and vegetables retailers in large city (like Dhaka) have higher reduction in sales and profit loss compared to retailers in small city (like Manikganj). These findings might be crucial for developing government support program for the retailers and others small and medium business owners.”

In PP20, Line 410 – 415 similar approaches followed

“The modelling exercise registered the importance of location of doing business for dietary foods. The varying severity of impact due to COVID is an empirical contribution to the theoretical field of research concerning “location model”[33]. Also, the findings has valid policy implication from practical perspective elucidating that for fiscal intervention location consideration as well as the consideration of product type is of vital importance to make the realistic impact on the group of interest.”

Comments on Conclusion: Writing and thoroughly outlining the policy implications in accordance with research in Bangladeshi contexts will provide more value.

Response to Comments on Conclusion:

Authors appreciated the observation and revised the conclusion section accordingly. In PP20-21 Line 428 – 436 the following is added

“While the country is experiencing an upward trend in the NCDs, any factors affecting the supply chain actors of healthy food is projected to degrade the situation. It was established that the location of the business (Dhaka or Manikganj) and type of business item (vegetables or fruits or both) are important while analyzing the reduction in profit margin and percentage change in sales. However, the type of retailer (WM or SS, or MV) is not significant. Thus, area-specific and product-specific interventions are required for the retailers of vegetables and fruits to minimize their vulnerability. Designing and implementing the such government interventions can result in price stability of dietary foods such as vegetables and fruits leading to the increase in their intake and help combating NCDs.”

Also in the discussion section in the revised version of the manuscript in PP20 Line 412 – 415 it now mentions

“Also, the findings has valid policy implication from practical perspective elucidating that for fiscal intervention location consideration as well as the consideration of product type is of vital importance to make the realistic impact on the group of interest.”

Comments on Tables and Figures: Tables and Figures need to reconfirmed according to the author’s guidelines of ‘PLoS ONE’.

Response to Comments on Tables and Figures:

Thanks for the observation. In the revised manuscript, all the tables and figures follow the guideline and requirements by PLOS ONE.

Comments on References: It is highly recommended that referencing styles should be reconfirmed according to the author’s guidelines of ‘PLoS ONE’.

Response to Comments on References:

The revised manuscript has a list of reference that follows the reference style followed and approved by PLOS ONE.

Comments on Overall or Others: This manuscript has met the quality requirements for publication in 'PLoS ONE' I wish the writers continued success in their efforts to improve the paper.

Response to Comments on Overall or Others:

Thanks, and gratitude to Reviewer 2 for the positive notes. The authors are encouraged as the reviewer have found that the manuscript has merit and quality for the Journal.

---

## [Decision Letter · Decision Letter 1]

22 Dec 2022

Vegetables and fruits retailers in two urban areas of Bangladesh: Disruption due to COVID – 19 and implications for NCDs

PONE-D-22-20793R1

Dear Dr. Abdullah S. M.,

We’re pleased to inform you that your manuscript has been judged scientifically suitable for publication and will be formally accepted for publication once it meets all outstanding technical requirements.

With Kind Regards,

Asst. Prof. Dr. Nemer Badwan

PhD in Economics and Finance

Assistant Professor of Economics and Finance

Academic Editor

PLOS ONE

Additional Editor Comments (optional): None.

Reviewers' comments:

Reviewer's Responses to Questions

**Comments to the Author**

1. If the authors have adequately addressed your comments raised in a previous round of review and you feel that this manuscript is now acceptable for publication, you may indicate that here to bypass the “Comments to the Author” section, enter your conflict of interest statement in the “Confidential to Editor” section, and submit your "Accept" recommendation.

Reviewer #2: All comments have been addressed

2. Is the manuscript technically sound, and do the data support the conclusions?

Reviewer #2: Yes

3. Has the statistical analysis been performed appropriately and rigorously? 

Reviewer #2: Yes

4. Have the authors made all data underlying the findings in their manuscript fully available?

Reviewer #2: Yes

5. Is the manuscript presented in an intelligible fashion and written in standard English?

Reviewer #2: Yes

6. Review Comments to the Author

Reviewer #2: After review the paper it has been identified that this manuscript has met the quality requirements for publication in 'PLoS ONE' I wish the writers continued success in their efforts to improve the paper.

7. PLOS authors have the option to publish the peer review history of their article (what does this mean?). If published, this will include your full peer review and any attached files.

Reviewer #2: **Yes: **Dr. Mohammad Delwar Hussain

Assistant Professor, Green Business School

Assistant Proctor, Permanent Campus

Green University of Bangladesh

Cell: +88-01891-460016

---

## [Editor Report · Acceptance letter]

2 Jan 2023

PONE-D-22-20793R1 

Vegetables and fruits retailers in two urban areas of Bangladesh: Disruption due to COVID – 19 and implications for NCDs 

Dear Dr. Abdullah:

I'm pleased to inform you that your manuscript has been deemed suitable for publication in PLOS ONE. Congratulations! Your manuscript is now with our production department. 

Kind regards, 

on behalf of

Asst. Prof. Dr. Nemer Badwan 

Academic Editor

PLOS ONE